# Grid-Connected Microbial Fuel Cell Modeling and Control in Distributed Generation

Fangmei Jiang [1,2], Liping Fan [3], Weimin Zhang [2] and Naitao Yang [2,*]

1 School of Information and Control, Shenyang Institute of Technology, Fushun 113122, China
2 School of Chemistry and Chemical Engineering, Shandong University of Technology, Zibo 255049, China
3 College of Information Engineering, Shenyang University of Chemical Technology, Shenyang 110142, China
* Correspondence: naitaoyang@126.com

**Abstract:** Water shortages and water pollution have seriously threatened the sustainable development of the community. The grid-connected microbial fuel cell is an effective way to control the cost of wastewater treatment plants. Moreover, it solves the problem of low efficiency and high energy consumption. In view of the characteristics of strong coupling, non-linearity, and internal load in the process of microbial fuel cell grid connection, it is necessary to design the grid-connected unit of power electronic device. Based on the establishment of the microbial fuel cell stack model, the stability control and the constant power control scheme were designed for the chopper and inverter, respectively. The simulation results showed that the control strategy with the combination of voltage stabilizer and constant power can make a grid-connected system of all phase voltage and frequency output. The three-phase voltage $U_{abc}$ was steady at 7 h and the voltage amplitude was controlled at roughly 380 V, according to the output voltage waveform. The value was 50 Hz, which satisfies the criteria for grid connection.

**Keywords:** microbial fuel cells; Boost converter; inverter; double loop control structure; constant power control

## 1. Introduction

The shortage of water resources and pollution of the water environment have become more and more serious, restricting the sustainable development of modern society [1,2]. With the fast development of industries, water pollution control has become the main way to achieve the harmonious coexistence of man and nature. As a centralized treatment unit for urban pollutants, sewage treatment plants occupy a very important position in the water environment pollution treatment strategy [3,4]. Nowadays, sewage release exhibits an increasing trend year by year, and the processing capacity of sewage treatment plants poses a severe challenge to power consumption [5]. Therefore, the development of renewable energy technology is critical because it provides a viable path for green production and environmentally friendly operation for the energy-saving operation of sewage treatment systems [6].

Distributed power generation (DG) energy, such as microbial fuel cells (MFC), has increasingly attracted extensive attention [7]. Typically, the microbial fuel cell is a device that uses pollutants and sludge in the sewage treatment process to directly convert chemical energy into electrical energy through the catalysis of specific microorganisms. The power generation process does not produce any harmful gases that pollute the environment. It is considered a promising renewable energy source for the future. One of its practical ways is to connect the microbial fuel cell to the load and energy storage equipment of the sewage treatment plant, forming a "microgrid" (MG). A complete local energy network can also be connected to the municipal power grid through a public connection point [8].

Generally, microgrid refers to the use of advanced control technology and power electronic devices to connect distributed energy sources with the loads and energy storage equipment by which it supplies to form a micro-integrated grid. The grid-connected microgrid can be connected to the municipal grid to achieve the purpose of reduced cost, increased reliability, and improved environmental performance of the sewage treatment plant [9,10]. Compared with the combination of multiple intermittent energy systems in traditional microgrid systems, microbial fuel cells can continuously provide a stable power output. Therefore, they enable the traditional microgrid systems to integrate with photovoltaics, wind energy, and secondary power sources [11]. The microgrid combined with the microbial fuel cell is an effective way to realize the transition from the traditional power grid to the smart grid by the active distribution network (ADN) [12]. The combination of microbial fuel cell and microgrid technology is equivalent to the combination of body and brain. For example, the microbial fuel cell stack provides reliable power output, while the microgrid system performs intelligent deployment control [13]. In addition, the output of the microbial fuel cell stack is DC power with a relatively large voltage variation range [14]. Its control system design should aim for constant voltage output to ensure the stability of the system; its inherent time-varying, uncertain and strong coupling, non-linearity, and other characteristics add difficulty to its control system design. In addition, the microgrid has characteristics of small inertia, strong randomness, and complex internal load. Its efficient operation depends on the control system design of the grid-connected unit composed of power electronic devices [15].

The process direction is the foundation of the majority of MFC research. There are not many control plans that are solely for MFC single cells with constant voltage output. In this research, using the constant voltage and constant power output of the grid-connected microbial fuel cell reactor as the control goal, we not only regulated the output voltage of single cell MFC, but also obtained a stable voltage which can be linked to the grid [16,17]. In order to design a control system for a grid-connected microbial fuel cell stack, it was necessary to create the corresponding control schemes according to the control objectives of different units [18]. In this paper, model predictive control (MPC) was used to design a constant pressure control scheme for the microbial fuel cell stack to achieve the stable operation of the subsystem. To address the problem of dynamic response delay in the main circuit of the grid, the voltage stabilization control of the chopper and the constant power control of the inverter under the double closed-loop structure were adopted, respectively [19]. The dual closed-loop PI control in the DC/DC chopper circuit can stabilize the voltage and respond quickly to the load. The constant power (PQ) control under the dual closed-loop structure in the DC/AC inverter circuit can realize effective active and reactive power tracking and improve the dynamic response speed of the system [20].

## 2. Modeling of the Microbial Fuel Cell Grid-Connected System

The electricity generation process of a microbial fuel cell can be represented by the following half-electrochemical reactions [21]:

Anode reaction:

$$(CH_2O)_2 + 2H_2O \rightarrow 2CO_2 + 8H^+ + 8e^- \tag{1}$$

Cathodic reaction:

$$O_2 + 4e^- + 2H_2O \rightarrow 4OH^- \tag{2}$$

The mathematical modeling of microbial fuel cells takes charge balance as the starting point. The charge balance of the anode and cathode, respectively, fit the following equations:

$$C_a \frac{d\eta_a}{dt} = 3600 i_{mfc} - 8Fr_1 \tag{3}$$

$$C_c \frac{d\eta_c}{dt} = -3600 i_{mfc} - 4Fr_2 \tag{4}$$

where $C_a$ and $C_c$, respectively, represent the capacitance of the anode and cathode. $\eta_a$ and $\eta_c$ are the overpotentials of the anode and cathode, respectively. $F$ is the Faraday constant; $i_{mfc}$ is the current density of a single cell. $r_1$ and $r_2$, respectively, represent the chemical reaction rates of the anode and the cathode reactions. Their values can be expressed by the following formulas:

$$r_1 = k_1^0 \exp\left(\frac{\alpha F}{RT}\eta_a\right)\frac{C_H}{K_H + C_H}X \tag{5}$$

$$r_2 = -k_2^0 \exp\left[(\beta - 1)\frac{F}{RT}\eta_C\right]\frac{C_{OH}}{K_{OH} + C_{OH}} \tag{6}$$

$k_1^0$ and $k_2^0$ are the anode and cathode reaction rate constants under standard conditions (maximum growth rate ratio); $\alpha$ and $\beta$ are the charge transfer coefficients of the anode and cathode reactions. $K_H$ and $K_{OH}$ are the half-reaction rate constants of hydrogen ion and hydroxide ion, respectively. $R$ is the gas constant. $T$ is stable operation. $C_H$ and $C_{OH}$ are the hydrogen ion concentration in the anode compartment and the hydroxide ion concentration in the cathode compartment. Their values can be calculated by the following formulae:

$$C_H = 10^{[-pH]} \tag{7}$$

$$C_{OH} = 10^{[pH-14]} \tag{8}$$

Assuming that the electrolytes in the anode and the cathode chambers of the microbial fuel cell are both continuously stirred, the mass balance equation of the anode chamber can be expressed as the follows [22,23]:

$$V_a\frac{dC_H}{dt} = Q_a\left(C_H^{in} - C_H\right) + 8A_m r_1 \tag{9}$$

$$V_a\frac{dX}{dt} = Q_a\frac{\left(X^{in} - X\right)}{f_X} + A_m Y_{ac} r_1 - V_a K_{dec} X \tag{10}$$

$$V_a\frac{dC_{AC}}{dt} = Q_a\left(C_{AC}^{in} - C_{AC}\right) - A_m r_1 \tag{11}$$

$$V_a\frac{dC_{CO_2}}{dt} = Q_a\left(C_{CO_2}^{in} - C_{CO_2}\right) + 2A_m r_1 \tag{12}$$

The mass balance equation of the cathode chamber can be expressed as follows:

$$V_c\frac{dC_{OH}}{dt} = Q_c(C_{OH}^{in} - C_{OH}) - 4A_m r_2 \tag{13}$$

$$V_c\frac{dC_{O_2}}{dt} = Q_c(C_{O_2}^{in} - C_{O_2}) + A_m r_2 \tag{14}$$

$$V_c\frac{dC_M}{dt} = Q_c(C_M^{in} - C_M) + A_m N_M \tag{15}$$

The subscripts '*in*', '*a*', and '*c*' stand for feed, anode, and cathode, respectively. $C_{AC}$, $C_{CO2}$, and X represent the concentrations of acetic acid, carbon dioxide, and biomass in the anode compartment, respectively. $C_{O2}$ and $C_M$ represent the dissolved oxygen and cation concentrations of the cathode compartment. $V$, $A_m$, and $Q$ represent the volume of the reactor, the cross-sectional area of the proton exchange membrane and the liquid flow rate, respectively. $f_x$, $Y_{ac}$, and $K_{dec}$ represent the reciprocal of the eluted fraction, bacterial equivalent, and acetic acid use delay constant. $N_M$ represents the cation flow from the anode chamber to the cathode chamber and its calculation formula can be expressed as follows:

$$N_M = \frac{3600 i_{mfc}}{F} \tag{16}$$

The output voltage $V_{mfc}$ of the microbial fuel cell can be calculated by the following formula:

$$V_{mfc} = V^0 - \eta_a + \eta_c - \left(\frac{d_{mfc}}{k^{aq}} + \frac{d^m}{k^m}\right)i_{mfc} \tag{17}$$

$V^0$ is the open circuit voltage of the fuel cell. $d_{mfc}$ and $d^m$ are the distance between the electrodes and the thickness of the proton exchange membrane, respectively; $k^{aq}$ and $k^m$ are the conductivity of the reaction solution and the proton exchange membrane, respectively [24,25].

Energy efficiency is produced by microbial fuel cells in the environment where they are used. The ratio of real energy to theoretical standard enthalpy change in oxidation conversion of anodic organic matter is the definition of energy efficiency. In actuality, it displays the proportion of organic matter's overall energy that is transformed to electric energy.

$$EE = \frac{E}{\Delta H} \times 100\% \tag{18}$$

E is short for electric energy J measured through the test (obtained by simulation waveform). The typical enthalpy change in the reaction between an electron donor and acceptor, as computed theoretically, is written as H stands for J.

$$E = P\Delta t \tag{19}$$

P is a representation of the battery's working power, w, and t is working time right now, s.

The correlation between Gibbs free energy and standard enthalpy change in thermodynamic calculations is calculated as follows:

$$\Delta H = -\Delta G^0 \tag{20}$$

Check the table after adding acetate content to the anode reaction formula to see how the standard enthral changes to $-487$ J, then = 487 J. E = 162 J represents the actual measured electric energy as seen in the waveform. 33% is the final energy efficiency.

The microbial fuel cell grid-connected system features a two-stage topology, which is mainly composed of a microbial fuel cell stack, Boost chopper, inverter, filter, circuit, and an AC power grid. The structure is shown in Figure 1. After the DC voltage output by the microbial fuel cell stack is boosted by the DC/DC chopper, the DC voltage is converted into AC voltage by the DC/AC inverter. The output voltage is satisfied by the LC filter [26].

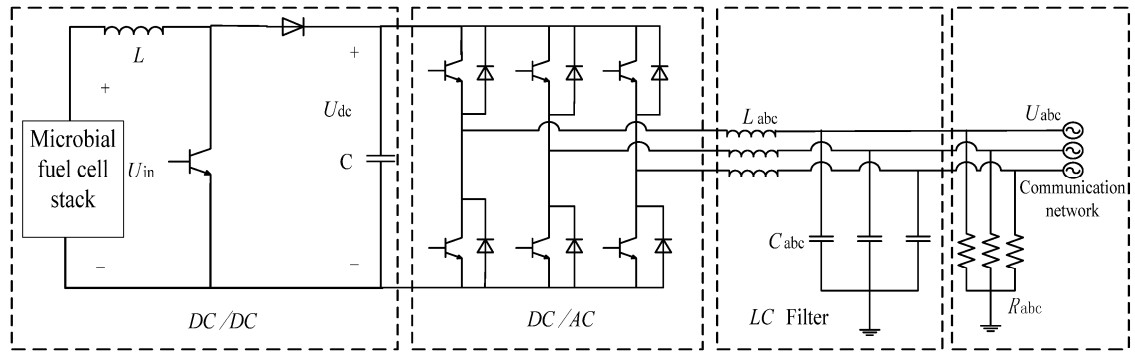

**Figure 1.** Grid-connected system structure diagram of two-stage microbial fuel cell stack.

### 2.1. Boost Chopper Modeling

DC/DC Converter is a single switch Boost Converter, which has excellent input/output performance and a simple circuit structure. In Figure 2, T is a controllable switch and R is a pure resistive load. When the switch T is switched on within $t_{on}$ time, the current $i_d$ flows through the load resistance R, and there is voltage $u_0$ at both ends of R. When the switch

T is disconnected at $t_{off}$ time, the current $i_0$ in R is zero, and the voltage $u_0$ becomes zero. Figure 3 shows the voltage and current wave-forms of the DC converter circuit [27,28].

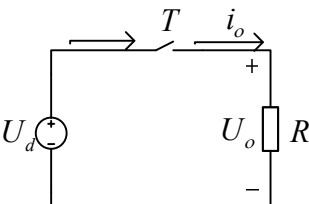

**Figure 2.** DC converter circuit.

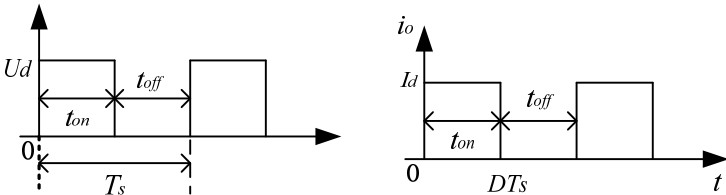

**Figure 3.** Waveform of voltage and current on load.

The duty cycle of the switches in the above circuit can be defined as follows:

$$D = \frac{t_{on}}{T_s} \tag{21}$$

where $T_s$ is the working period of switch $T$, and $t_{on}$ is the on-off time of switch $T$. The average output voltage can be obtained from the waveform diagram.

$$U_o = \int_0^{T_s} u_d dt = \frac{t_{on}}{T_s} U_d = DU_d \tag{22}$$

If the switch is considered to have no loss, the output power is as follows:

$$P_o = \frac{1}{T_s} \int_0^{DT_s} u_o i_o dt = D\frac{U_d{}^2}{R} \tag{23}$$

In the formula, $u_d$ is the input DC voltage, because $D$ is the coefficient of change between 0 and 1. Therefore, the average output voltage $u_0$ is always less than the input DC voltage $u_d$ within the range of $D$, and changing the $D$ value can change the value of the average output voltage. The change of duty cycle can be achieved by changing $t_{on}$ or $T_s$.

The DC/DC converter used in a system for microbiological fuel cells should have the ability to control the output voltage of the MFC, and its performance should satisfy the following two criteria:

(1) As a component that conveys energy to other components, DC/DC converters have high conversion efficiencies, which increase energy usage rates;

(2) The grid-connected microbial fuel cell reactor has a high output voltage demand. The converter should feature a voltage boost capability to lower the need.

Therefore, the Boost type DC/DC converter is preferred for microbial fuel cell systems, and its basic form is shown in Figure 4.

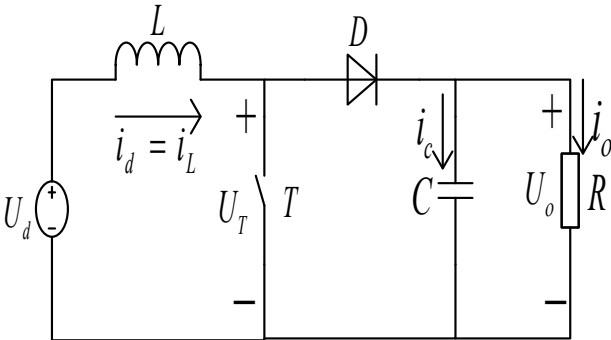

**Figure 4.** Boost type conversion circuit.

Given that switch T drives the signal, when the circuit is turned on, the switch is in the on state during the working time, the diode is under reverse bias voltage, and the diode has cut-off frequency. Inductance L stores energy that is supplied by the DC power source, and the inductance current $i_L$ rises linearly as a result. Capacitor C, by contrast, powers load R [29,30].

The circuit is at $t_{off}$; during the working time of the switch off state, switch T is used to regulate the off signal. The diode D is turned on. Since the current in the inductance L cannot be mutated, the opposite inductive electromotive force is generated to prevent the current from decreasing [31]. We can see the DC/DC conversion circuit in the grid-connected microbial fuel cell reactor as shown in Figure 5.

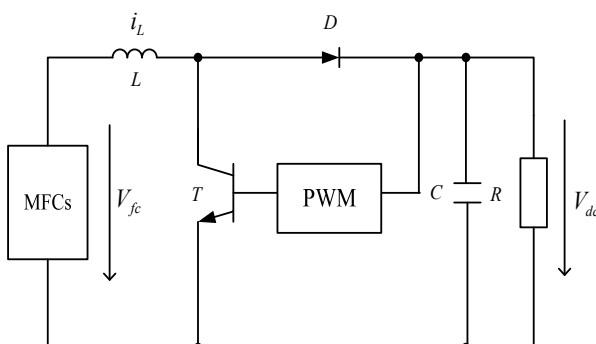

**Figure 5.** DC/DC converter circuit in grid-connected microbial fuel cell reactor.

The DC/DC chopper utilizes the boost circuit structure, as shown in Figure 1. When the inductance L and the capacitance C are large enough, they can be controlled only by the switching of "on" and "off" [32]. The switching period is [t, t+] and the duty cycle is D (0 < D < 1). In the switching period [t, t+], the steady-state model of the Boost chopper is as follows:

$$U_{dc} = \frac{1}{1-D} U_{in} I_{dc} = (1-D) I_{in} \tag{24}$$

(1) The time interval for the switching on is [t, t+]. Here, the DC power supply charges the inductance L through the switching device, the inductance stores energy, and the capacitor C releases energy and supplies power to the subsequent circuit [33]. The formulas are as follows:

$$L \frac{dI_{in}}{dt} = U_{L(on)} \tag{25}$$

$$C \frac{dU_{dc}}{dt} = i_{c(on)} \tag{26}$$

(2) When the switching device is in the off state, the time interval is [t+, t+]. The microbial fuel cell power supply and the inductance $L$ charge the capacitor C together and supply power to the subsequent circuit [34]. The expression is as the follows:

$$L\frac{\mathrm{d}I_{\mathrm{in}}}{\mathrm{dt}} = U_{\mathrm{in}} - U_{\mathrm{dc}} \tag{27}$$

$$C\frac{\mathrm{d}U_{\mathrm{dc}}}{\mathrm{dt}} = I_{\mathrm{dc}} - I_{\mathrm{in}} = i_{\mathrm{c(off)}} \tag{28}$$

### 2.2. Inverter Modeling

The DC/AC inverter adopts a three-phase voltage type inverter circuit, and its structure is shown in Figure 6. The three-phase voltage inverter uses capacitors to store energy on the DC side, while the DC side presents a low-impedance voltage source characteristic. The power of an ideal controlled inverter should remain unchanged after the tuning of DC to AC, and the DC voltage waveform and current waveform should not produce pulsation [35]. Its structure is shown in Figure 7.

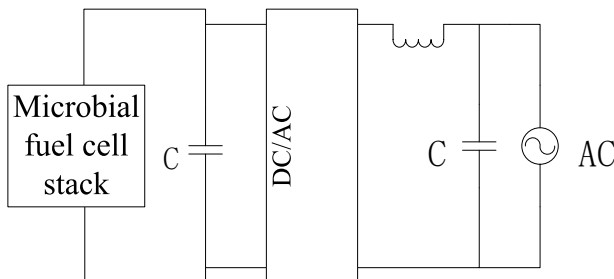

**Figure 6.** Three-phase voltage type inverter circuit.

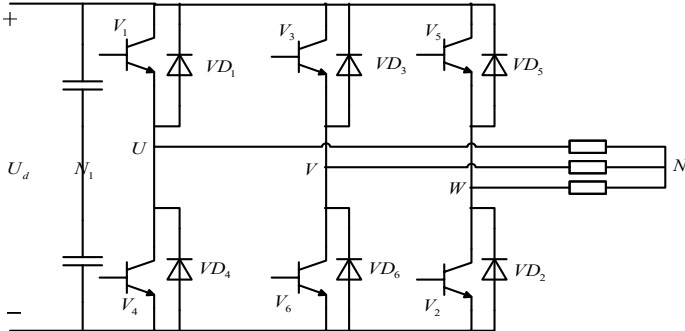

**Figure 7.** Inverter structure diagram.

The basic working mode of the three-phase voltage bridge inverter circuit is 180 degrees. The conduction angle of each bridge arm is 180 degrees. The longitudinal commutation mode is adopted for the same phase, and the upper and lower bridge arms conduct electricity alternately. Within one cycle, the conduction of the 6 conduction trigger tubes was performed in the order of $T_1{\rightarrow}T_2{\rightarrow}T_3{\rightarrow}T_4{\rightarrow}T_5{\rightarrow}T_6$, and the difference of the 6 conduction trigger tubes was 60 degrees successively. In addition, three trigger tubes were simultaneously on at any one time. Conduction in the 6 groups followed $T_1T_2T_3$, $T_2T_3T_4$, $T_3T_4T_5$, $T_4T_5T_6$, $T_5T_6T_1$, and $T_6T_1T_2$, each composite conducting 60 degrees [36,37].

Assuming the load three-phase symmetry, we can obtain:

$$u_{UN} + u_{VN} + u_{WN} = 0 \tag{29}$$

$$u_{NN_1} = \frac{1}{3}(u_{UN_1} + u_{VN_1} + u_{WN_1}) \tag{30}$$

That is, the $u_{NN1}$ waveform is a rectangular wave, the frequency is 3 times that of $u_{NN1}$, and the amplitude is 1/3, namely $u_d/6$.

*2.3. Filter Modeling*

The filter adopts an LC filter circuit, and its transfer function expression is as follows:

$$\frac{U_{\text{out}}(S)}{U_{\text{in}}(S)} = \frac{1}{\omega_L^2 S^2 + \frac{2\xi}{\omega_L} S + 1} \tag{31}$$

$\omega_L$ is the LC resonance angular frequency. $\xi$ is the damping coefficient, and its expression is as follows:

$$\omega_L = \frac{1}{\sqrt{LC}} \tag{32}$$

$$\xi = \frac{\sqrt{L}}{2R\sqrt{C}} \tag{33}$$

The cut-off frequency of the LC filter can be expressed by the following formula:

$$f_L = \frac{1}{\left(2\pi\sqrt{LC}\right)} \tag{34}$$

The selected area is $10f_1 < f_L < f_s/10$. It is the fundamental frequency and the carrier frequency of SPWM.

Based on the above mathematical models, a simulation operation platform of grid-connected microbial fuel cell stack was established under the MATLAB/Simulink simulation environment, and the simulation operation experiment of the control scheme of each subsystem was carried out on the basis of this simulation platform.

## 3. Design of Control System for Grid-Connected Microbial Fuel Cell Stack

Figure 8 shows a flow chart of the control system of the grid-connected microbial fuel cell stack. Based on the establishment of the mechanism model of the microbial fuel cell stack, model predictive control was used to design a constant voltage control system for MFCs to ensure the stability of the output voltage of MFCs [38]. Then, the voltage stabilization control and constant power control schemes under the dual closed-loop PID control structure were designed for the chopper and inverter, respectively. Finally, the output voltage met the grid-connected demand through LC filtering [39].

Based on the dual-stage grid-connected power generation system structure, the Boost chopper adopts a dual closed-loop PID control structure, and its structure is shown in Figure 9. The outer loop PI controller tracks the preset outer loop reference voltage and generates the inner loop reference current; the inner loop PI controller tracks the reference current to generate the pulse signal D that controls the switching device, and finally realizes the improvement of the response speed of the chopper circuit and the control target of steady-state accuracy [40].

The three-phase voltage inverter adopts a double closed-loop structure to design a constant power (PQ) control scheme, and its control goal is to make the active and reactive power output by the DC/AC inverter meet the expected goals at the same time. That is, when the voltage and frequency of the AC network change within the specified range, the active and reactive power output by the inverter remain unchanged.

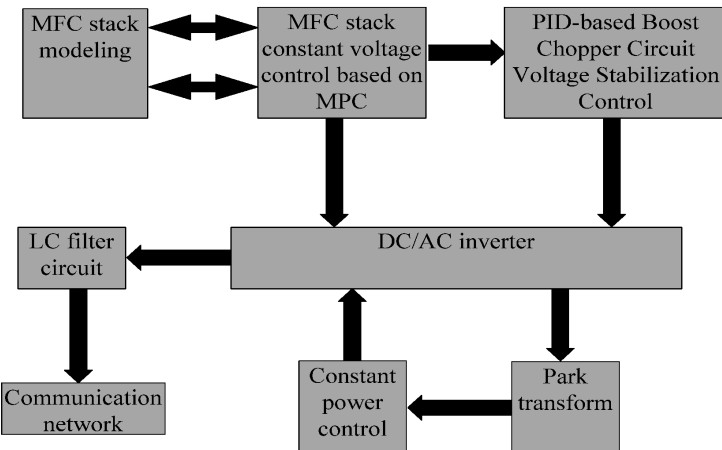

**Figure 8.** Flow chart of grid-connected microbial fuel cell stack control system.

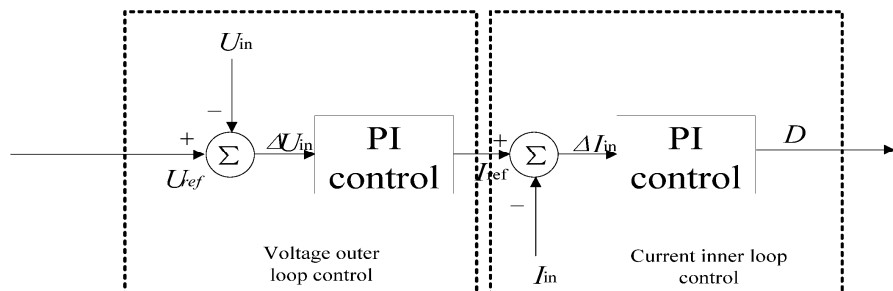

**Figure 9.** Boost chopper double closed-loop control system structure.

When the voltage and frequency of the connected power grid change within the predetermined range, constant power regulation maintains the output power of the distributed power supply at the reference value. Decoupling active power from reactive power is accomplished through constant power regulation. The distributed power system's initial operational state has a frequency of $f_0$ and a voltage of $U_0$ at the connecting bus. In this instance, the reference values for the active power and reactive power, $P_{ref}$ and $Q_{ref}$, are the output active power and reactive power, respectively. Both the reactive power controller and the active power controller can modify their characteristic curves for voltage and frequency changes [41].

When the frequency and voltage vary within a reasonable range, i.e., $f_{min} \leq f \leq f_{max}$, $U_{min} \leq U \leq U_{max}$, the active power and reactive power, or the reference value $P_{ref}$ and $Q_{ref}$, remain constant. The voltage and frequency cannot be kept in a stable state using this control strategy. As a result, a stable control voltage and frequency controller are required for operation on an isolated island [42].

To obtain the dq axis components ($i_{dq}$ and $u_{dq}$) and the instantaneous power $P_{grid}$ and $Q_{grid}$, first take the three-phase instantaneous current $i_{abc}$ and instantaneous voltage $u_{abc}$, then park transform them. To acquire the average power $P_{filt}$ and $Q_{filt}$, the obtained instantaneous power is put through a low-pass filter. It is compared to the reference power $P_{ref}$ and $Q_{ref}$ that is provided, its error is controlled using PID, and the reference current signal $I_{dref}$ and $I_{qref}$ of the inner loop controller are output. For tracking static errors, PID is used. The error is set to zero until the controller is stable when the inverter's output power is equal to the reference power [43]. We can see that the typical structure of a constant power outer loop controller is shown in Figure 10.

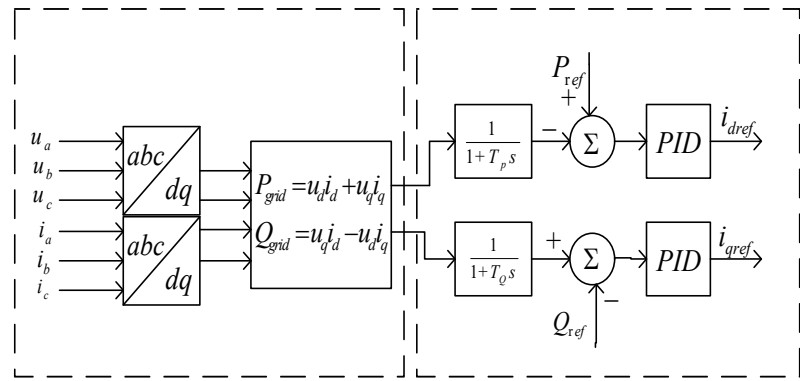

**Figure 10.** Typical structure of constant power outer loop controller.

$P_{grid}$ and $Q_{grid}$ are instantaneous power, as illustrated by the following formulae:

$$P_{grid} = \frac{U_1 U}{Z} cos(\varphi_Z - \varnothing) - \frac{U^2}{Z} cos\varphi_Z \tag{35}$$

$$Q_{grid} = \frac{U_1 U}{Z} sin(\varphi_Z - \varnothing) - \frac{U^2}{Z} sin\varphi_z \tag{36}$$

$Z$ represents the reactance between the inverter and the AC network; $\varphi_Z$ stands for line impedance angle; $U$ represents the amplitude of the voltage on the AC network side; $U_i$ represents the amplitude of the output voltage at the inverter side; and $\varphi$ represents the phase difference between the inverter voltage and the AC network voltage.

Given $P_{ref}$ and $Q_{ref}$, the output voltage amplitude and phase angle reference values $U_{Iref}$ and $\varphi_{Iref}$ of the inverter are obtained as follows:

$$U_{Iref} = \sqrt{\frac{Z^2}{U^2}(P_{ref}^2 + Q_{ref}^2) + U^2 + 2P_{ref}Z \cos \varphi_Z + 2Q_{ref}Z \sin \varphi_Z} \tag{37}$$

$$\varphi_{Iref} = \varphi_Z - \arccos(\frac{ZP_{ref}}{UU_{Iref}} + \frac{U}{U_{Iref}} \cos \varphi_Z) \tag{38}$$

The constant power control scheme first decouples the instantaneous power into active and reactive parts. The instantaneous active power $P_{grid}$ and instantaneous reactive power $Q_{grid}$ are obtained by recombining the $i_d$, $i_q$, $u_d$ and $u_q$ obtained after park transformation of the three-phase instantaneous current $i_{abc}$ and instantaneous voltage $u_{abc}$. In the double closed-loop structure, through the $P_{grid}$ and $Q_{grid}$ comparison with the active reference voltage $P_{ref}$ of the outer loop and the reactive reference voltage $Q_{ref}$ of the inner loop, the error is PID controlled to obtain the reference signal $i_{dref}$ and $i_{qref}$ of the inner loop controller. The inner loop control adopts dq$_0$ rotating coordinate system control to improve power quality and system operation performance. The constant power control scheme can be expressed by the following formulae:

$$i_{dref} = \frac{P_{ref}}{u_d} \tag{39}$$

$$i_{qref} = -\frac{P_{ref}}{u_d} \tag{40}$$

$$u_{Fd} = u_d + Ri_d + L\prime\frac{di_d}{dt} - \omega L\prime i_q \tag{41}$$

$$u_{Fq} = Ri_q + L'\frac{di_q}{dt} + \omega L' i_d \tag{42}$$

Its inverter constant power (PQ) control system structure is shown in Figure 11:

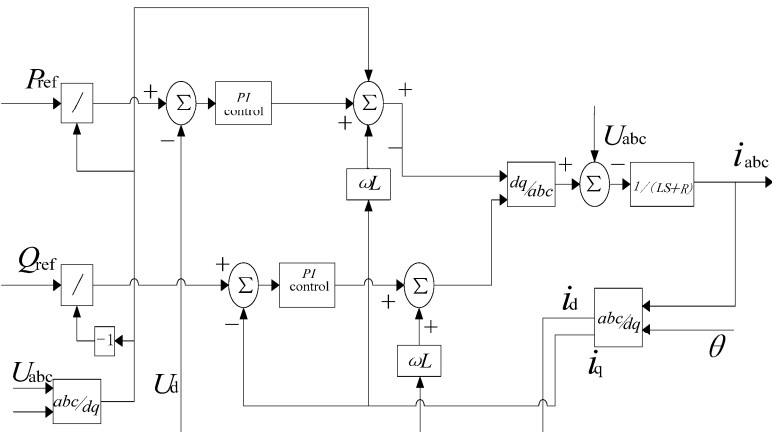

**Figure 11.** Inverter constant power (PQ) control system structure.

## 4. Simulation Run

With the microbial fuel cell reactor as the core of the microgrid, the microbial fuel cell reactor was modeled using simulink. Through MPC control, the output waveform of the voltage and power of the microbial fuel cell reactor was obtained. The output voltage and output power of the microbial fuel cell stack are shown in Figure 12.

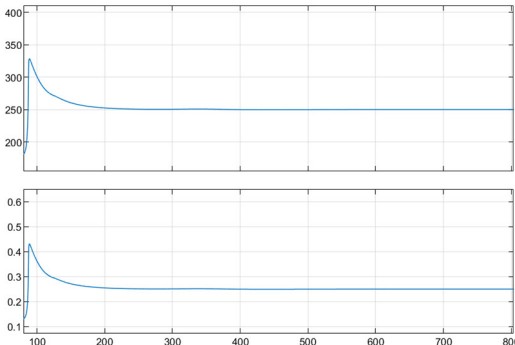

**Figure 12.** Output voltage and output power of microbial fuel cell stack.

In order to verify the effectiveness of the control scheme designed for each subsystem and the overall output characteristics of the combined subsystems, the overall output performance of the control system was verified through simulation analysis in the MATLAB/Simulink simulation environment [44,45]. Based on the modeling of each subsystem, the inverters were compared and simulated using PWM control and PQ control. The parameters used in the simulation are shown in the following Table 1:

Figures 13 and 14 are the simulation results of grid-connected voltage under traditional pulse width modulation (PWM) control and constant power (PQ) control schemes, respectively. It can be seen from the simulation results that under the condition of traditional pulse width modulation (PWM) control, the output voltage of the system has obvious harmonics and poor stability, which cannot meet the grid connection requirements. When the constant power (PQ) control scheme is adopted, the voltages of each phase are output stably under the same frequency conditions, the voltage harmonics meet the expected targets, and all performance indicators meet the grid-connected requirements.

**Table 1.** Parameters used in simulation.

| | |
|---|---|
| $U_{stack}$/V. | 675 |
| Grid frequency $f_n$/$H_Z$ | 50 |
| Carrier frequency $f_s$/$H_Z$ | 4000 |
| Filter inductance $L_f$/mH | 10 |
| Filter capacitor $C_f$/mF | 10 |
| Filter resistance $R_f$/$\Omega$ | 0.16 |
| Line resistance R $_\Omega$/Km | 0.641 |
| Line inductance X $_\Omega$/Km | 0.101 |
| Load/MW | 1 |
| DC/DC:PID control parameter | $K_P = 10$ |
| | $K_i = 1$ |
| | $K_d = 0.5$ |
| DC/AC: PQ control parameter | $K_{pP} = 0.15$ |
| | $K_{pQ} = 1.154$ |
| | $K_{iP} = 0.15$ |
| | $K_{iQ} = 1.154$ |
| Power voltage/V | 380 |
| Capacity/MVA | 40 |
| $p_{ref}$/Kw | 12 |
| $Q_{ref}$/Var | 0 |

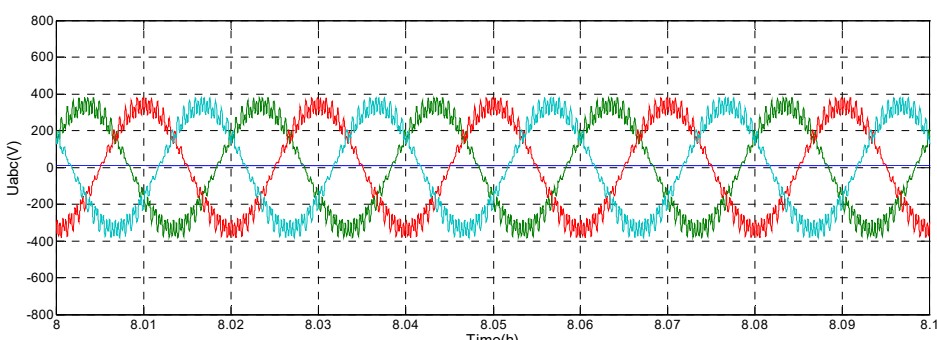

**Figure 13.** Grid−connected voltage under pulse width modulation (PWM) control.

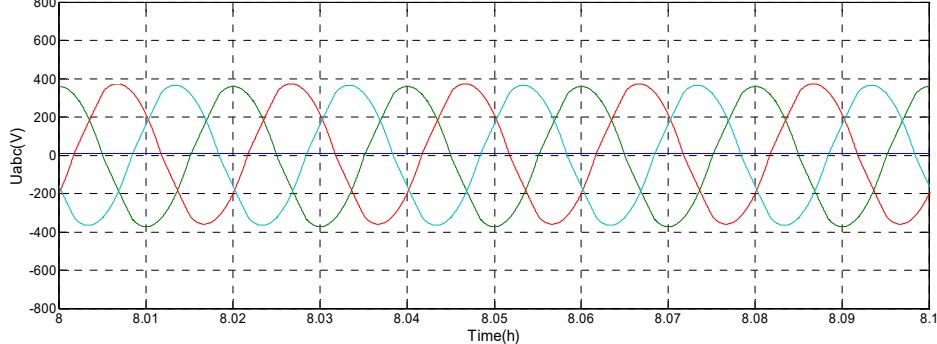

**Figure 14.** Grid−connected voltage under constant power control.

## 5. Conclusions

It can be determined that PQ control considerably reduces the ripple when compared to the output of PWM control and PQ control. The output three-phase voltage can support a big grid connection and is more stable. The active power and reactive power remain within the reference range when the voltage and frequency of the large power grid fluctuate within the designated range. The huge power grid must keep its voltage and frequency stable for PQ control.

The grid-connected microbial fuel cell stack has the characteristics of strong coupling, nonlinearity, and complex internal load. Its efficient operation depends on a well-designed control scheme for the grid-connected unit composed of power electronic devices [46,47]. Using a constant power control scheme to improve the performance of the inverter in the grid-connected unit can weaken the drawbacks caused by pulse width modulation (PWM) and improve the dynamic and steady-state performance of the three-phase output voltage of the system. The control system design of the grid-connected unit of the microbial fuel cell stack provides a theoretical basis and technical support for the efficient and energy-saving operation of the sewage treatment process, which has important academic and practical significance.

The MFC reactor model itself produces limited electricity, requires a lot of single batteries to build, is difficult to construct, and requires a lot of maintenance to stay stable. Although theoretical research has shown that its inverter's control precision can satisfy the requirements of real-world industrial processes, there are still certain technical issues with the hardware implementation stage that prevent the control accuracy from having the desired impact [48,49]. In order to reach the goals of the industrialist, it is going to be necessary to further resolve the current issues in the field.

**Author Contributions:** Methodology, W.Z.; Writing—original draft, F.J.; Supervision, L.F. and N.Y. All authors have read and agreed to the published version of the manuscript.

**Funding:** This research received no external funding.

**Data Availability Statement:** The data presented in this study are available on request from the corresponding author. The data are not publicly available due to the model belongs to the individual.

**Conflicts of Interest:** The authors declare no conflict of interest.

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
