# Peer review of "Grid-Connected Microbial Fuel Cell Modeling and Control in Distributed Generation"

_processes, doi:10.3390/pr11020466_

Round 1

Reviewer 1 Report

Brief Summary:

This paper proposes a control system for a grid-connected MFC stack. The simulation results show that the voltages of each phase are output stably under constant power control schemes, the voltage harmonics meet the expected targets, and all performance indicators meet the requirement of grid connection.

I felt confident that the authors performed careful simulation processing, and the data obtained is convincing. However, some problems must be solved before it is considered for publication. I explain my concerns in more detail below.

Major Comments:

1.     The main concern I have about the paper is the insufficient background and relevant references. What are the shortcomings of previous studies on the control system for a microgrid?

2.     It is better to conduct some bench-scale study besides simulation.

3.     Equation (1): Acetic acid is concerned to be the only electron donor. There is no problem with simplifying when modeling, but this is not practical for a MFC, especially in sewage treatment plants.

4.     Equation (8): superscript “14-pH” should be “pH-14”.

Minor Comments:

5.     It is recommended to use acronyms after your first definition.

6.     Line 23: “seriously”, misused.

7.     Line 50: “be able to”, remove the “be”

8.     Check other grammatical errors.

Author Response

  1. The main concern I have about the paper is the insufficient background and relevant references. What are the shortcomings of previous studies on the control system for a microgrid?

Response:Thank you for the nice reminders. Several obstacles still exist for microbial fuel cells: (1) Battery structure issue: In order to use microbial fuel cells commercially on a wide scale, it is important to further simplify the battery structure, decrease the volume and weight, enhance the cost-performance ratio, and lower the cost difference with conventional power production. (2) Output voltage quality issue: A single microbial fuel cell produces low-voltage direct current as its output voltage. The battery stack should be constructed in a specific fashion and then employed following the converter boost treatment in order to accommodate the requirements of various loads. The output voltage of a microbial fuel cell will alter in accordance with how the external load varies step-wise. Therefore, to enhance the quality of the voltage output and satisfy the needs of various loads, a good automatic control device is required.

  1. It is better to conduct some bench-scale study besides simulation.

Response:Thank you for the nice reminders.The laboratory experiment study is being carried out step by step since the system involves several links and is constrained by environmental restrictions and financial resources, which is also the direction of our efforts. Complete experimental simulation cannot be carried out at this time.

  1. Equation (1): Acetic acid is concerned to be the only electron donor. There is no problem with simplifying when modeling, but this is not practical for a MFC, especially in sewage treatment plants.

Response:Thank you for the nice reminders.A two-compartment microbial fuel cell reactor with an anode chamber and a cathode chamber makes up an MFC unit. The pre-treated electrode and proton exchange membrane were installed in the appropriate positions. The wastewater accaccated sludge was placed into the bottom of the anode reaction chamber as a microbial strain, and then the prepared anode liquid and cathode liquid were injected into the anode chamber and the cathode chamber, respectively, to start the MFC and enter the operation state.Similar to Model 1 of anaerobic digestion, the mathematical model of MFC was developed (ADM1).

CH stands for the anode's acetate concentration, and X for the biomass content. Thus, the equation that we are modeling is this one.

  1. Equation (8): superscript “14-pH” should be “pH-14”.

Response:Thanks for your kind reminders.It has been corrected.

Minor Comments:

  1. It is recommended to use acronyms after your first definition.

Response:Thanks for your kind reminders.It has been corrected.

  1. Line 23: “seriously”, misused.

Response:We are really sorry for our mistakes.Thank you for your reminder.It has been corrected.

  1. Line 50: “be able to”, remove the “be”

Response:We are really sorry for our mistakes.Thank you for your reminder.It has been corrected.

  1. Check other grammatical errors.

Response:We are really sorry for our mistakes.Thank you for your reminder.We'll check it out.

Reviewer 2 Report

This work could be reconsidered after addressing carefully the following comments:

1.The title of the manuscript should be more sharp.

2.The abbreviations should not be used in the title, abstract, and keywords.

3.The motivation and contribution of this works must be clarified in the introduction section.

4.Some quantities results should be presented in abstract and conclusion.

6.The ideas/criteria of choosing the size/arrangement of considered system should be given?

7.More explanation is required in results and discussion section.

8.For more contribution, the authors should compare their results with the related results in other published works.

9.There are many grammatical errors and typoerros throughout the whole manuscript? The paper should be rechecked.

10.The novelty of the work must be clearly addressed and discussed, compare your research with existing research findings and highlight novelty, (compare your work with existing research findings and highlight novelty).

11.Conclusion: Future scope of the work should be provided.

12.The literature review section is very weak. As they are many recent published papers on the same topic need to be included.

Author Response

This work could be reconsidered after addressing carefully the following comments:c n    

  1. The title of the manuscript should be more sharp.

Response:Thanks for your reminders.Because the purpose of this work is primarily to reflect modeling and grid-connection control, we have made the following changes.Grid-connected Microbial Fuel Cell Modeling and Control in Distributed Generation.

  1. The abbreviations should not be used in the title, abstract, and keywords.

Response:Thanks for your kind reminders.We refer to changes in PQ control as constant power control.

  1. The motivation and contribution of this works must be clarified in the introduction section.

Response:Thanks for your kind reminders.We have added a sentence to the introduction section(line68 to line78),to clarify this.

The process direction is the foundation of the majority of MFC research. There aren't many control plans that are solely for MFC single cells with constant voltage output. In this research, using the constant voltage and constant power output of the grid-connected microbial fuel cell reactor as the control goal, we not only regulate the output voltage of single cell MFC, but also obtain a stable voltage which can be linked to the grid.

  1. Some quantities results should be presented in abstract and conclusion.

Response:thanks for your kind reminders.We have added a sentence to the introduction section(line18 to line20),to clarify this.

The three-phase voltage Uabc is steady at 7h, and the voltage amplitude is controlled at roughly 380V, according to the output voltage waveform. The Hz value is 50. satisfies the criteria for grid connection.

  1. The ideas/criteria of choosing the size/arrangement of considered system should be given?

Response:thanks for your kind reminders.A two-compartment microbial fuel cell reactor, an external load system, and a data collecting system make up the MFC experimental system. The cathode chamber, anode chamber, proton exchange membrane, cathode, and anode make up the majority of the two-compartment MFC. The proton exchange membrane between the two reaction chambers has a surface area of 5×10-4 m2, the anode electrode and the cathode electrode are made of carbon cloth, and the MFC's anode reaction chamber and cathode reaction chamber have a combined volume of 5.5×10-5 m3. However, this is simply a model for an experimental single-microbial fuel cell.According to the generated parameter table, which is only used for the simulation model's parameter design, a high number and size of fuel cell stacks are actually needed as a grid-connected power source.

  1. More explanation is required in results and discussion section.

Response:thanks for your kind reminders.We have added a sentence to the introduction section(line299 to line304),to clarify this.

It can be determined that PQ control considerably reduces the ripple when compared to the output of PWM control and PQ control. The output three-phase voltage can support a big grid connection and is more stable. PQ command The active power and reactive power remain within the reference range when the voltage and frequency of the large power grid fluctuate within the designated range. The huge power grid must keep its voltage and frequency stable for PQ control.

  1. For more contribution, the authors should compare their results with the related results in other published works.

Response:Thanks for your reminder.I have read the pertinent literature, and the primary findings concern the microbial fuel cells themselves and managing the voltage of a single cell or a battery stack. There hasn't been much research linking microbial fuel cells to grid-connected systems. In this essay, I wish to compare the outcomes of my two control strategies, PWM control and continuous power control.

8.There are many grammatical errors and typoerros throughout the whole manuscript? The paper should be rechecked.

Response:Thanks for your reminder. we will check and revise the overall grammar of the draft.

  1. The novelty of the work must be clearly addressed and discussed, compare your research with existing research findings and highlight novelty, (compare your work with existing research findings and highlight novelty).

Response:Thanks for your reminder.The majority of the microbial fuel cell study, according to the research literature, is focused on the single cell's design, control, and other elements. The focus of this work is the grid-connected control system, which is addressed in addition to controlling the voltage of a single battery.

11.Conclusion: Future scope of the work should be provided.

Response:Thanks for your reminder.we have added a sentence to the conclusion section(line315 to line321)to describe future scope.

The MFC reactor model itself produces limited electricity, requires a lot of single batteries to build, is difficult to construct, and requires a lot of maintenance to stay stable. Although theoretical research has shown that its inverter's control precision can satisfy the requirements of real-world industrial processes, there are still certain technical issues with the hardware implementation stage that prevent the control accuracy from having the desired impact. In order to reach the goal of industrialisation, it's going to be necessary to further resolve the current issues in the field.

  1. The literature review section is very weak. As they are many recent published papers on the same topic need to be included.

Response:Response:thanks for your kind reminders.We enriched the references.

Reviewer 3 Report

My review comments are below:

1. This paper has insufficient novelty since simple DC/DC, DC/AC and filter blocks have been cascaded to connect the source to the grid. The authors should explain the novelty of this study in detail. I found that a simple control scheme is described that is connected to converters to make the power flow from source to the grid. Therefore I suggest authors investigate more references and make a wide comparison between different DC/DC and inverter converters and their controllers to show the reaction of the proposed converter for these blocks. Below you can find sample references for DC/DC side (a-d) and inverter side (e-h):

1.a: Ghaderi, D.; Maroti, P.K.; Sanjeevikumar, P.; Holm-Nielsen, J.B.; Hossain, E.; Nayyar, A. A Modified Step-Up Converter with Small Signal Analysis-Based Controller for Renewable Resource Applications. Appl. Sci. 2020, 10, 102. https://doi.org/10.3390/app10010102

1.b: Qun Qi, Davood Ghaderi, Josep M. Guerrero, Sliding mode controller-based switched-capacitor-based high DC gain and low voltage stress DC-DC boost converter for photovoltaic applications, International Journal of Electrical Power & Energy Systems, Volume 125, 2021, 106496, ISSN 0142-0615, https://doi.org/10.1016/j.ijepes.2020.106496.

1.c:S. Padmanaban, N. Priyadarshi, M. S. Bhaskar, J. B. Holm-Nielsen, E. Hossain and F. Azam, "A Hybrid Photovoltaic-Fuel Cell for Grid Integration With Jaya-Based Maximum Power Point Tracking: Experimental Performance Evaluation," in IEEE Access, vol. 7, pp. 82978-82990, 2019, doi: 10.1109/ACCESS.2019.2924264.

1.d: W. Wang et al., "Power Decoupling Control for Single-Phase Grid-Tied PEMFC Systems With Virtual-Vector-Based MPC," in IEEE Access, vol. 9, pp. 55132-55143, 2021, doi: 10.1109/ACCESS.2021.3071776.

1.e: doi: 10.1109/TIE.2018.2804898.

1.f: https://doi.org/10.1002/2050-7038.12300

1.g: https://doi.org/10.1016/j.compeleceng.2020.106575

1.h: DOI: 10.1109/TSTE.2017.2785738

The authors should cite these references and some other newly published papers in MDPI, IEEE, Elsevier,.. etc journals to show the impact of their proposed controller.

2. The efficiency of this topology under the proposed controller reaction is not presented. Both the mathematical investigations and graphical presentations are advised to be presented.

3. Section 3  should be explained in more detail. The proposed equations are not enough and it is hard to understand how the control scheme is drawn. 

Author Response

  1. This paper has insufficient novelty since simple DC/DC, DC/AC and filter blocks have been cascaded to connect the source to the grid. The authors should explain the novelty of this study in detail. I found that a simple control scheme is described that is connected to converters to make the power flow from source to the grid. Therefore I suggest authors investigate more references and make a wide comparison between different DC/DC and inverter converters and their controllers to show the reaction of the proposed converter for these blocks. Below you can find sample references for DC/DC side (a-d) and inverter side (e-h):

1.a: Ghaderi, D.; Maroti, P.K.; Sanjeevikumar, P.; Holm-Nielsen, J.B.; Hossain, E.; Nayyar, A. A Modified Step-Up Converter with Small Signal Analysis-Based Controller for Renewable Resource Applications. Appl. Sci. 2020, 10, 102. https://doi.org/10.3390/app10010102

1.b: Qun Qi, Davood Ghaderi, Josep M. Guerrero, Sliding mode controller-based switched-capacitor-based high DC gain and low voltage stress DC-DC boost converter for photovoltaic applications, International Journal of Electrical Power & Energy Systems, Volume 125, 2021, 106496, ISSN 0142-0615, https://doi.org/10.1016/j.ijepes.2020.106496.

 1.c:S. Padmanaban, N. Priyadarshi, M. S. Bhaskar, J. B. Holm-Nielsen, E. Hossain and F. Azam, "A Hybrid Photovoltaic-Fuel Cell for Grid Integration With Jaya-Based Maximum Power Point Tracking: Experimental Performance Evaluation," in IEEE Access, vol. 7, pp. 82978-82990, 2019, doi: 10.1109/ACCESS.2019.2924264.

 1.d: W. Wang et al., "Power Decoupling Control for Single-Phase Grid-Tied PEMFC Systems With Virtual-Vector-Based MPC," in IEEE Access, vol. 9, pp. 55132-55143, 2021, doi: 10.1109/ACCESS.2021.3071776.

 1.e: doi: 10.1109/TIE.2018.2804898.

 1.f: https://doi.org/10.1002/2050-7038.12300

1.g: https://doi.org/10.1016/j.compeleceng.2020.106575

 1.h: DOI: 10.1109/TSTE.2017.2785738

 The authors should cite these references and some other newly published papers in MDPI, IEEE, Elsevier,.. etc journals to show the impact of their proposed controller.

Response: Thanks for your reminding. I thoroughly studied the materials offered, and I can really relate to the concepts you gave me here. The goal of choosing between boost chopper modeling and inverter modeling is to select an appropriate control strategy for the inverter model. V/F control and droop control are also chosen among the three control methods. The best outcome is achieved by the PQ control technique. Hence PQ control is chosen. The other two control systems, however, are not described owing to space issues. However, the suggested research direction is likewise deserving of my careful consideration and investigation, and it is also what I need to focus on next. Thank you.

  1. The efficiency of this topology under the proposed controller reaction is not presented. Both the mathematical investigations and graphical presentations are advised to be presented.

Response: I appreciate your reminders. In terms of efficiency, I didn't monitor its maximum power because my goal was to get to the necessary voltage of 380V for grid connection. The resulting stack voltage for a single battery powered at 0.5V is 500V. The graph below displays the sole voltage produced by a single battery.

But I'll keep researching efficiency and incorporate the power tracking module. Many thanks

  1. Section 3  should be explained in more detail. The proposed equations are not enough and it is hard to understand how the control scheme is drawn. 

Response:thanks for your reminder, We have added a sentence to the introduction section(line222 to line252),to clarify this.                               

Round 2

Reviewer 2 Report

This work could be accepted for publication in the present modified version.

Author Response

Response:Many thanks for your appreciation.

Reviewer 3 Report

Some of the corrections have been done, however, my review comments have not been considered (neither suggested references and similar studies, nor efficiency calculations) in the revised paper. The novelty of this work still is not presented and can not be published in the present state.

Author Response

Response:I apologize if the most recent change did not fulfill your requirements. I changed the following things this time. I now have a better knowledge of the BOOST circuit thanks to the material you suggested. I utilize a double closed-loop PID to regulate the boost circuit and have a better grasp of the inverter's operating principle. The microbial fuel cell stack's output voltage and power are also increased. The process through which microorganisms use the acetic acid in sewage to break down and produce electricity is the mechanism model for the microbial fuel system. To observe the system's conversion efficiency, the energy efficiency calculation is also included. Once again, thank you for the article and its concepts.

Round 3

Reviewer 3 Report

A big part of my questions has been addressed.